# Perceived stress and coping strategies among nursing students towards rejoining college after covid-19 pandemic

**Sarmila Koirala**[1] *, **Rupa Devi Thapa**[2], **Sagun Bhandari**[2], **Alisha Rijal**[2], **Bhawani Shahi Thakuri**[3], **Pooja Gauro**[2]

**1** Nepalese Army Institute of Health Sciences, College of Nursing, Sanobharyang, Bhandarkhal, Kathmandu, **2** Yeti Health Science Academy, Kantimarga, Maharajgunj, Kathmandu, **3** Kathmandu Model Hospital, School of Nursing, Swayambhu, Kathmandu

* sarmilakoirala45@gmail.com

**Data Availability Statement:** All relevant data are within the manuscript and its Supporting Information files.

## Abstract

The COVID-19 pandemic has had a significant impact on nursingeducation, with many programs abruptly discontinuing clinical practice. This has raised concerns among nursing students about their clinicalskill development. Because of this, the researcher was motivated to learn how nursing students perceive stress and develop coping mechanisms for returning to nursing colleges following the COVID-19 pandemic. This study aimed to assess perceived stress and coping strategies among nursing students toward rejoining college after COVID -19 pandemic.Ananalyticalcross-sectionalstudywas conducted using proportionatestratified simple random sampling technique among 317 nursing students of bachelor level at all nursing colleges affiliated to Purbanchal University in Kathmandu Valley. Data was collected usingaself-administeredquestionnaire that included the Perceived Stress Scale (PSS) and the Brief COPEScale.The majority of respondents (71.3%) reported moderate stress levels, with 28.7%reportinghighstresslevels.Half of the respondents (51.7%) had low coping levels, while the other half (48.3%) had high coping levels. There was a significant association between the level of coping and mother's occupation (p = 0.003). The Pearson correlation between perceived stress and coping strategies was moderately positive (r = 0.256, p = 0.001).Nursing college administration and instructors can play a role in reducing student stress and promoting coping strategies by maintaining a safecollege environment for students rejoining college after the COVID-19 pandemic.

## Introduction

COVID-19 was declared as pandemic by the World Health Organization (WHO) on March 11, 2020. The disease is caused by the new coronavirus SARS-CoV-2, which quickly spread across the World. The virus was identified initially in Wuhan, China, in December 2019 and was defined in early January. Public Health Emergency of International Concern was announced on January 30, 2020. COVID-19's devastating and unpredictable global spread has

**Funding:** This work was supported by the University Grants Commission Nepal (https://www.ugcnepal.edu.np/) under Grant No. SRIDG-77/78, awarded to SK, SB, and BST. The funders had no role in study design, data collection and analysis, decision to publish, or preparation of the manuscript.

**Competing interests:** The authors have declared that no competing interests exist.

resulted in global lockdowns and a massive burden on healthcare systems [1]. The first COVID-19 case in Nepal was discovered on January 23, 2020, in a 65-year-old Nepali student who hadjust returned from Wuhan, China [2]. A 19-year-old woman who had returned from France on March 17 was diagnosed with COVID-19 on March23, 2020 [3]. Like other countries Nepal also locked its international borders, instituted a statewide lockdown, and stopped all schools to stop sickness in its earliest stages till June 14, 2020.After the lockdown, colleges and schools continued education by distance education. Youngsters started worrying about COVID-19, the duration of the pandemic, social distancing, which caused them to experience stress [4]. Many universities implemented preventive measures, including closing colleges, canceling classes, transitioning to online-based teaching/learning, examinations and postponing practical based learning. No one could predict when or whether they will be able to resume their activities as before. Such disruptions can exert extra additional pressures which adversely affect students' mental health, increasing stress, anxiety, and depression [5,6]. Considering the pandemic's magnitude and rapid spread, the increased worry and stress in the general population, students, and healthcare workers are understandable [7,8]. The disease significantly impacts mental health, causing people to experience various degrees of emotional problems [9]. College students' mental health and unhealthy behavior patterns took important place in professional and public discussions [10].

It can be noticed that individuals' mental illness is an economic burden, especially for their families, both in terms of treatment costs and reducing productivity [11]. Low or moderate stress levels might enhance students' motivation, leading to greater effort while studying and achieving their goals. Conversely, high-stress levels can negatively influence students, leading to anxiety and depression, affecting students' health as well as academic performance [12].The interruption of education for nursing students has been unexpected from students. In addition, the clinical practice of nursing students in hospitals has also been discontinued. Because much of nursing education consists of clinical practice, students may have been concerned about inadequate clinical skill development. Also, the application skills are insufficient, so the uncertainty of when, where and how to do the compensatory practicum to eliminate inadequacy could stress nursing students [13]. Along with this students were scared to resume college and their practicum due to spread and emergence of new variants.

In the US 71% of students reported increased stress due to COVID-19 [14]. In Spain, approximately half of the students, 47.92% experienced a moderate level of stress [15]. Likewise, Turkish nursing students experienced a moderate level of stress, with a perceived stress score of 31.69 ± 6.91 [14]. Saudi, nursing students experience a high-stress level [16]. Similarly, Indian nursing students showed a moderate level of stress during COVID-19, with a maximum mean perceived stress score 22.56 [17]. In Nepal, 84.1% of nursing students perceived a moderate stress level [18]. Coping mechanisms are also essential to deal with nursing students' daily stress. Longitudinal studies have shown that stress levels in nursing students may increase or decrease during their educational training, depending on their coping behavior strategies. Besides these differences, coping strategies vary according to the individual's characteristics and the context where the stressors are found [19]. Problem-solving strategies have been identified as one of the best ways to cope with stress. Conversely, emotional-based coping strategies appear to be the least influential [20]. Nepalese nursing students had a moderate coping strategy (63.3%), and adaptive coping strategies were highly used [19]. However, the direction of the relationship between emotional responses and coping strategies is not clear, and the connection is not always constant [21]. The alarming prevalence of 80% of perceived stress among medical college students, including nursing, the use of different coping strategies, and their recommendation for the need to stress management programs [22] contribute to the importance of research in this issue.

As a nursing teacher, the researcher observed stressful concerns among many nursing students and their parents regarding the completion of the academic year. The stress is both on theoretical and practical course completion. Currently, students are attending online classes, which are gradually becoming their new normal. They express not having the will to resume physical classes and clinical posting due to fear of COVID-19 infection and getting habituated to online courses over time. Due to the mandatory rule for fulfilling the practical requirement of nursing education, the students must rejoin physical classes and clinical postings sooner or later. Therefore the researcher is interested and aimed to find out the level of perceived stress, its association with selected variables, identifycoping strategies adopted by nursing students towards rejoining college, its association with selected variables and determine the relationship between perceived stress and coping strategies adopted by students after the COVID-19 pandemic. This will provide evidence to make informed strategies to facilitate enabling environment to motivate students to rejoin physical classes and clinical postings.The study might help nursing college administration and instructors identify stress among students and formulate the protocol for coping strategies which might ultimately improve the quality of nursing education and produce productive students. The findings of the study might be used as a reference point for further research in the future.

## 1.1 Research questions

Is there perceived stress among nursing students towards rejoining college after COVID-19 pandemic?

What coping strategies are adopted by nursing students towards rejoining college after COVID-19 pandemic?

## 1.2 Objectives of the study

**1.2.1 General objective.** To find out nursing students' perceived stress and coping strategies towards rejoining college after the COVID-19 pandemic.

**1.2.2 Specific objectives.** To identify the level of perceived stress among nursing students towards rejoining college.

To identify the level of coping strategies nursing students adopt towards rejoining college.

To measure the association between perceived stress levels among nursing students and selected variables.

To measure the association between the level of coping strategies adopted by nursing students and selected variables.

To determine the relationship between perceived stress and coping strategies adopted by students.

## 1.3 Operational definition

**Nursing students**: It refers to students studying in 2nd, 3rd and 4th Year Bachelor of science in Nursing (BSN)as well as 2nd and 3rdyear in Post Basic Bachelor of Nursing Science (PBNS).

**Perceived Stress**: Perceived stress refers to feelings or thoughts which causes emotional or physical tension among nursing students. It was measured by using Perceived Stress Scale (PSS).

**Level of stress:** The level of stress was categorized into 3 levels.

Low level- Scores ranging from 0–13.

Moderate level- Scores ranging from 14–26.

High level- Scores ranging from 27–40 [23].

**Coping strategies**: Coping strategies refers to the specific efforts, both behavioral and psychological, that the nursing students are employing to master, tolerate, reduce, or minimize

stressful events during COVID-19. It was measured by Brief-COPE Scale [24]. Coping strategies consists of:

**Problem focused coping**: It is characterised by the facets of active coping, use of informational support, planning, and positive reframing. A high score indicates coping strategies that are aimed at changing the stressful situation. High scores are indicative of psychological strength, grit, a practical approach to problem solving and is predictive of positive outcomes.

**Emotion focused coping**: It is characterised by the facets of venting, use of emotional support, humour, acceptance, self-blame, and religion. A high score indicates coping strategies that are aiming to regulate emotions associated with the stressful situation. High or low scores are not uniformly associated with psychological health or ill health, but can be used to inform a wider formulation of the respondent's coping styles.

**Avoidant coping**: It is characterised by the facets of self-distraction, denial, substance use, and behavioural disengagement. A high score indicate physical or cognitive efforts to disengage from the stressor. Low scores are typically indicative of adaptive coping.

**Level of coping**: It was classified into two categories according to mean score.

High Coping: Score > Mean (65.5)

Low Coping: Score ≤ Mean

**College**: It refers to those educational institutions which provide education and training to become a qualified nurse. It includes 11 nursing colleges affiliated to Purbanchal University and located at Kathmandu where both PBNS and BSN Nursing courses are being taught.

**Rejoining college**: It refers to the activity of nursing students of being physically present in college for continuing education after lockdown during COVID-19 pandemic.

**Type of family:** It refers to the family in which respondent is residing. It can be nuclear, joint and extended.

**Occupation:** It refers to the occupation in which respondents father and mother are engaged in for income generation. It can be agriculture, sales/services, business, professional job (government/ private)

**Education:** It refers to the level of education that respondents father and mother has obtained. It can range from unable to read and write, able to read and write but received no formal education, taken primary education (class 1–8), secondary(class 8–10), higher secondary(class 11–12), bachelors and masters.

**Income**: It refers to the total monthly income of respondent's family. It includes the income of father and mother both.

## 2. Research methods

### 2.1 study design

A descriptive, analytical study design was used to assess perceived stress and coping strategies among nursing students towards rejoining college after COVID -19 pandemic.

### 2.2 Study setting and population

This study was conducted in all nursing colleges having Bachelor of Science in Nursing (BSN) and Post Basic Bachelor of Nursing Science (PBNS) Programs affiliated to Purbanchal University located at Kathmandu Valley. There were altogether 11 nursing colleges which have both BSN and PBNS program in Kathmandu Valley.The population of the study was the bachelor level nursing students. The total study population was (N = 1476).Students studying in 2nd and 3rdyear of PBNS and students of 2nd,3rd and 4thyear of BSN program in nursing colleges located at Kathmandu Valley.

## 2.3 Sampling

Probability, proportionate stratified sampling technique was used for the study.Firstlyall nursing colleges having Bachelor of Science in Nursing (BSN) and Post Basic Bachelor of Nursing Science (PBNS) Programs affiliated to Purbanchal University located at Kathmandu Valley were selected purposively. Each college was taken as strata and sample from each strata was selected on proportionate basis..Sample size wascalculated by using Cochran's formula.

The Cochran formula is;

$$\text{Sample size}(n°) = Z^2pq/d^2.$$

Where Z = 1.96 at 95% level of significance
p (Prevalence of perceived stress) = 35.9% = 0.359 [25].
q = 1-p = 1–0.359 = 0.641, d = Sample error i.e. 5% = 0.05, N = Target population = 1476

$$\text{By formula, sample size}(n°) = \frac{(1.96)^2 \times 0.359 \times 0.641}{0.05 \times 0.05} = \frac{3.8416 \times 0.359 \times 0.641}{0.0025}$$

$$= \frac{0.95694256}{0.0025} = 353.61$$

Since population is finite

$$\text{Sample size}(n) = n°/1 + (n° - 1)/N$$
$$= 353.61/1 + (353.61 - 1)/1476$$
$$= 287.48$$

For 10% (non- respondent) = 287.48+28.7 = 317
Total sample size was 317.
Proportionate stratified sampling was used to select the sample from 11 nursing colleges.
n = 317
After identifying number of sample from each strata sample are drawn randomly by lottery method

## 2.4 Research instrument

Semi structured questionnaire developed. Standard tool, Perceived Stress Scale (PSS) developed by Cohen, Kamarck, &Mermelstein (1994) [23] was used to assess perceived stress among nursing students and Brief-COPE Scale developed by Carver (1997) [24]was used to find out the coping strategies adapted by nursing students.This instrument was divided into three parts: Part I- Questions related to socio-demographic and profession related information.Part II - Perceived Stress Scale (PSS). It is 5 point Likert scale ranging from 0–4, consisting 10 items with maximum possible score of 40.Part III- Brief-COPE Scale. It is 4 point Likert scale ranging from1-4 with 14 sub scales. Each subscale consists of 2 items where total items are 28 with maximum possible score of 112. It will take about 30 minutes to fill up the questionnaire.

## 2.5 Ethical considerations

Firstly, permission letter was taken from all 11 colleges for data collection and ethical approval was taken from Institutional Review Board (IRB) of Nepal Health Research Council for ethical clearance. Approval Number for this study is 657/2021P and Reference no 1302. Written

| S.N. | Name of College | Total population | Calculation | Required Sample |
|---|---|---|---|---|
| 1. | Asian College for Advanced Studies | BN (38+39 = 77) BSC (20+20+20 = 60) = 137 | 317/1476*137 | 30 |
| 2. | B&B Medical Institute | BN (36+40 = 76) BSC (20+20+20 = 60) = 136 | 317/1476*136 | 29 |
| 3. | Yeti Health Science Academy | BN (40+39 = 79) BSC (20+19+19 = 58) = 137 | 317/1476*137 | 30 |
| 4. | Kantipur Academy of Health Science | BN (38+38 = 76) BSC (20+20+18 = 58) = 134 | 317/1476*134 | 29 |
| 5. | Hope International College | BN (33+21 = 54) BSC (20+20+20 = 60) = 114 | 317/1476*114 | 24 |
| 6. | Norvic Institute of Nursing Education | BN (38+39 = 77) BSC (30+29+30 = 89) = 166 | 317/1476*166 | 35 |
| 7. | Everest College of Nursing | BN (36) BSC (20+20 = 40) = 76 | 317/1476*76 | 16 |
| 8. | Kathmandu Model College | BN (39+40 = 79) BSC (20+19+19 = 58) = 137 | 317/1476*137 | 30 |
| 9. | Om Health Campus | BN (40+39 = 79) BSC (30+30+30 = 90) = 169 | 317/1476*169 | 36 |
| 10. | Nepal Institute of Health Science | BN (38+39 = 77) BSC (20+20+18 = 58) = 135 | 317/1476*135 | 29 |
| 11. | Nagarik College of Health Science | BN (40+38 = 78) BSC (18+19+20 = 57) = 135 | 317/1476*135 | 29 |
| **Total** | | **BN (709)+BSC(767) = 1476** | | **317** |

informed consent was taken from each respondent after explaining the purpose of the study. Confidentiality of the information of the respondents was maintained by not disclosing the information and by using information for the purpose of the study. Respondents dignity was maintained by giving right to reject or discontinue from the study at any time without any penalty.

## 2.6 Data collection procedure

Researcher themselves collected the data. Firstlyrespondents were requested to sign informed consent by explaining the purposes of study. Data was collected through self administered questionnaire form which tookabout 30 minutes.Data was collected from 21st December 2021 to 11th September 2022. At the end of the data collection, questionnaires were checked for its completeness and accuracy.

## 2.7 Data analysis

The data was edited, organized, coded and entered in Statistical Package for Social Science (SPSS) version 20.Data was analyzed by using descriptive and appropriate inferential statistics that is chi-square test, Pearson correlation and presented in academic tables.

## 3. Results and discussion

The results are presented in two parts. Part I includes descriptive analysis and part II includes inferential analysis.

### Part I

Table 1 reveals the socio-demographic information of the respondents. The respondent's age ranged from 19–36 years with the mean age of 22.9±2.8. Majorityof the respondents (74.1%) belonged to age group 20–24 years. Most of the respondents' ethnicity belonged to Brahmin/Chettri (53.9%) and Janajati (42%) respectively. Majority of the respondents were hindu (81.4%). Almost 83.2% of the respondents were unmarried and most of them lived in home (80.8%) and 73.2% belonged to nuclear family.

Table 2 states the family related information of the respondents. Almost half (49.5%) of the respondent's monthly family income was above NRs, 50000. Approximately 27.8% of the father's education was higher secondary whereas 28.7% of the mother's education was secondary level. Almost half the respondent's fathers did some kind of business where as 67.2% of the mother's occupation was home maker.

**Table 1. Socio-demographic information of the respondents n = 317.**

| Variables | Frequency | Percentage(%) |
|---|---|---|
| **Age Group** | | |
| ≤ 19 years | 7 | 2.2 |
| 20–24 years | 235 | 74.1 |
| 25–29 | 65 | 20.5 |
| 30–34 | 9 | 2.8 |
| ≥ 35 years | 1 | 0.3 |
| *Mean± SD = 22.9 ±2.8Min = 19 years,Max = 36 years* | | |
| **Ethnicity** | | |
| Dalit | 4 | 1.3 |
| Janajati | 133 | 42 |
| Madhesi | 7 | 2.2 |
| Brahmin/Chettri | 171 | 53.9 |
| Others | 2 | 0.6 |
| **Religion** | | |
| Hindu | 258 | 81.4 |
| Buddhist | 45 | 14.2 |
| Christian | 12 | 3.8 |
| Others | 2 | 0.6 |
| **Marital Status** | | |
| Married | 53 | 16.7 |
| Unmarried | 264 | 83.2 |
| **Living Arrangements** | | |
| Home | 256 | 80.8 |
| Hostel | 26 | 8.2 |
| Rent and with Relatives | 35 | 11 |
| **Type of Family** | | |
| Joint | 70 | 21.1 |
| Nuclear | 232 | 73.2 |
| Extended | 15 | 4.7 |

**Table 2. Family related information of the respondents n = 317.**

| Variables | Frequency | Percentage(%) |
|---|---|---|
| **Family Income** | | |
| Below 20,000 | 15 | 4.7 |
| 20,001–30,000 | 35 | 11 |
| 30,001–40,000 | 45 | 14.2 |
| 40,001–50,000 | 65 | 20.5 |
| 50,001 and above | 157 | 49.5 |
| **Father's Education** | | |
| Unable to read and write | 2 | 0.6 |
| Can just read and write | 18 | 5.7 |
| Primary level | 41 | 12.9 |
| Secondary level | 92 | 29 |
| Higher Secondary | 88 | 27.8 |
| Bachelor | 72 | 22.7 |
| Masters and above | 4 | 1.3 |
| **Mother's Education** | | |
| Unable to read and write | 17 | 5.4 |
| Can just read and write | 56 | 17.7 |
| Primary level | 63 | 19.9 |
| Secondary level | 91 | 28.7 |
| Higher Secondary | 60 | 18.9 |
| Bachelor | 27 | 8.5 |
| Masters and above | 3 | 0.9 |
| **Occupation of Father** | | |
| Agriculture | 39 | 12.3 |
| Sales &services/ professional | 97 | 30.6 |
| Business | 165 | 52.1 |
| Others | 16 | 5 |
| **Occupation of Mother** | | |
| Home Maker | 213 | 67.2 |
| Agriculture | 23 | 7.3 |
| Sales &services/ professional | 74 | 23.3 |
| Other's | 7 | 2.2 |

Table 3 states the professional characteristics of the respondents. More than half (56.8%) of the respondents were studying Bachelor of Science in Nursing. Near about half of the them (47.3) studied in third year. Almost 32.2% of them were involved in work activities. None of them were exposed in any kinds of pandemic before. More than half of the respondent received some kind of support during pandemic.

Table 4 reveals that the perceived stress score ranged from 14 to 40 with a mean of 22.95 ($SD$ = 8.09). The perceived stress scale's absolute skew value was lower than 2 and absolute kurtosis value lower than 7. The brief coping score ranged from 28–105 with a Mean±SD of 65.55±*12.45*. The scores for problem focused ranged from 8 to 32 with Mean±SD of 21.69± 4.62, emotion focusedranged from 12 to 47with Mean±SD of 27.9± 6.29and avoidant coping ranged from 8 to 29with Mean±SD of 15.95± 4.05 which states that during stress student mostly adopted problem focused coping strategies with mean percentage of 67.78.

Table 5, reveals the level of perceived stress among the respondents. Majority (71.3%) of the respondents has moderate and almost one third (28.7%) of the respondents has high stress.

**Table 3. Professional characteristics related information of respondents n = 317.**

| Variables | Frequency | Percentage(%) |
|---|---|---|
| **Academic Institution** | | |
| Kantipur Academy of Health Sciences | 29 | 9.1 |
| Asian College for Advanced Studies | 30 | 9.5 |
| B&B Medical Institute | 29 | 9.1 |
| Nepal Institute of Health Sciences | 29 | 9.1 |
| Hope International College | 24 | 7.6 |
| Nagarik College of Health Science | 29 | 9.1 |
| Om Health Campus | 36 | 11.4 |
| Norvic Institute of Nursing Education | 35 | 11 |
| Everest College of Nursing | 16 | 5 |
| Yeti Health Science Academy | 30 | 9.5 |
| Kathmandu Model College | 30 | 9.5 |
| **Program** | | |
| PBBN | 137 | 43.2 |
| BSc. Nursing | 180 | 56.8 |
| **Academic Year** | | |
| Second | 88 | 27.8 |
| Third | 150 | 47.3 |
| Fourth | 79 | 24.9 |
| **Work during Studies** | | |
| Yes | 102 | 32.2 |
| No | 215 | 67.8 |
| **Previous Exposure to Pandemic** | 317 | 100 |
| No | | |
| **Received Support during Pandemic** | | |
| Yes | 168 | 53 |
| No | 149 | 47 |

Table 6, reveals the level of coping among the respondents. More than half of the respondents (51.7%) have low coping and near about half (48.3%) of the respondents have high coping.

## Part II

Table 7 shows that there is no significant association between level of stressand socio-demographic information of the respondents.

Table 8 shows that there was no significant association between level of stress and family related information of the respondents.

**Table 4. Descriptive analysis of perceived stress scale scores and brief coping scale scores of the respondents n = 317.**

| Characteristics | Minimum to Maximum Scores | | | | Obtained Scores | | | | | |
|---|---|---|---|---|---|---|---|---|---|---|
| | | Score Range | $\bar{x}$ | SD | Mean% | Skewness | SE of Skewness | $Z_{Skewness}$ | Kurtosis | SE of Kurtosis | $Z_{Kurtosis}$ |
| **Perceived Stress** | 0–40 | 14–40 | 22.95 | 8.09 | 57.37 | 1.120 | 0.137 | 0.38 | -0.184 | 0.273 | -0.67 |
| **Brief Coping** | 28–112 | 28–105 | 65.55 | 12.45 | 58.52 | 0.145 | 0.137 | 1.05 | 0.718 | 0.273 | 2.63 |
| Problem Focused | 8–32 | 8–32 | 21.69 | 4.62 | 67.78 | -0.295 | 0.137 | -2.15 | -0.275 | 0.273 | -1.07 |
| Emotion Focused | 12–48 | 12–47 | 27.90 | 6.29 | 58.32 | 0.335 | 0.137 | 2.44 | 0.354 | 0.273 | 0.20 |
| Avoidant | 8–32 | 8–29 | 15.95 | 4.05 | 49.84 | 0.667 | 0.137 | 4.86 | 0.422 | 0.273 | 1.54 |

**Table 5. Level of perceived stress of the respondents n = 317.**

| Level of Perceived Stress | Frequency | Percentage(%) |
|---|---|---|
| Moderate Stress | 226 | 71.3 |
| High Stress | 91 | 28.7 |
| **Total** | **317** | **100** |

**Table 6. Level of coping of the respondents n = 317.**

| Level of Coping | Frequency | Percentage(%) |
|---|---|---|
| Low coping | 164 | 51.7 |
| High coping | 153 | 48.3 |
| **Total** | **317** | **100** |

**Table 7. Association between level of stress and socio-demographic information of respondents n = 317.**

| Variables | Level of Stress | | $\chi^2$ | p-value |
|---|---|---|---|---|
| | Moderate No. (%) | High No. (%) | | |
| **Age group (in years)** | | | | |
| <20 | 26 (72.2) | 10(27.8) | 0.152 | 0.927* |
| 21–25 | 169 (71.6) | 67(28.4) | | |
| ≥26 | 31 (68.9) | 14(31.1) | | |
| **Ethnicity** | | | 0.235 | 0.889* |
| Janjati | 94 (70.7) | 39 (29.3) | | |
| Brahmin/Chettri | 122 (71.3) | 49 (28.9) | | |
| Others | 10 (76.9) | 3 (23.1) | | |
| **Religion** | | | | |
| Hindu | 179 (69.4) | 79 (30.6) | 2.480 | 0.115 |
| Other than Hindu | 47 (79.7) | 12 (20.3) | | |
| **Marital Status** | | | | |
| Married | 35 (66) | 18 (34) | 0.859 | 0.354 |
| Unmarried | 191 (72.3) | 73(27.7) | | |
| **Living Arrangements** | | | | |
| Home | 186 (72.7) | 70 (27.3) | 1.207 | 0.272 |
| Other than home | 40 (65.6) | 21 (34.4) | | |
| **Family Type** | | | | |
| Nuclear Family | 164 (70.7) | 68 (29.3) | 0.154 | 0.695 |
| Other than nuclear | 62 (72.9) | 23 (27.1) | | |

*Significant level at 0.05 * Likelihood Ratio.*

**Table 8. Association between level of stress and family related information n = 317.**

| Variables | Level of Stress | | $\chi^2$ | p-value |
|---|---|---|---|---|
| | Moderate No. (%) | High No. (%) | | |
| **Income** | | | | |
| 25000 and below | 41 (82) | 9 (18) | 3.69 | 0.158 |
| 25001 to 50000 | 74 (67.3) | 36 (32.7) | | |
| 50001 and above | 111 (70.7) | 46 (29.3) | | |
| **Father's Education** | | | 1.421 | 0.491 |
| Primary level and below | 42 (68.9) | 19 (31.1) | | |
| Secondary and higher secondary | 133 (73.9) | 47 (26.1) | | |
| Bachelor and above | 51 (67.1) | 25 (32.9) | | |
| **Mother's Education** | | | | |
| Primary level and below | 98 (72.1) | 38 (27.9) | 0.357 | 0.837 |
| Secondary and higher secondary | 108 (71.5) | 43 (28.5) | | |
| Bachelor and above | 20 (66.7) | 10 (33.3) | | |
| **Father's Occupation** | | | | |
| Agriculture | 28 (71.8) | 11 (28.2) | 2.045 | 0.563* |
| Sales and service/professional | 68 (70.1) | 29 (29.9) | | |
| Business | 121 (73.3) | 44 (26.7) | | |
| Others | 9 (56.3) | 7 (43.8) | | |
| **Mother's Occupation** | | | | |
| Homemaker | 157 (73.7) | 56 (26.3) | 2.142 | 0.543* |
| Agriculture | 15 (65.2) | 8 (34.8) | | |
| Sales and servicel | 50 (67.6) | 24 (32.4) | | |
| Others | 4 (57.1) | 3 42.9) | | |

*Significant Level at 0.05 * Likelihood ratio.*

Table 9 shows that there was no significant association between level of coping and socio-demographic variables of the respondents.

Table 10 shows that there was no significant association between level of coping and respondents' family income, parents' education and father's occupation but there was significant association between level of coping and mother's occupation (p = 0.003).

Table 11 depicts the pearson correlation which was calculated to find out bivariate relationship between perceived stress and coping strategies, significant relationship was found between stress and coping among respondents (r = 0.256, p = 0.001). The strength of relationship was moderately positive.

## Discussion

COVID-19 pandemic sudden onset had disrupted the life of peoples including students in many ways. The pandemic has altered how universities around the world operate. A sudden lockdown in Nepal led to a shift from traditional classroom settings to online learning, impacting university students' academic stress, anxiety, helplessness, and altered quality of life. Student's academic and personal lives have significantly changed due to lockdown, limitations, social isolation, quarantine, and hygiene. The most impacted sector by the COVID-19 pandemic is education. Students were anxious to return to school after the pandemic because of the development of numerous new COVID virus variants.

The range of the perceived stress score in this study was 14–40, with a mean of 22.95±8.09, which is comparable to the Spanish study's mean score of 22.78±8.54 [16], but different from

**Table 9. Association between level of coping and socio-demographic information n = 317.**

| Variables | Level of Coping Strategies | | χ² | p-value |
|---|---|---|---|---|
| | Low<br>No. (%) | High<br>No. (%) | | |
| **Age group (in years)** | | | | |
| <20 | 18 (50) | 18 (50) | 0.327 | 0.849 |
| 21–25 | 121 (51.3) | 115 (48.7) | | |
| ≥26 | 25 (55.6) | 20 (44.4) | | |
| **Ethnicity** | | | 2.868 | 0.238 |
| Janjati | 73 (54.9) | 60 (45.1) | | |
| Brahmin/Chettri | 87 (50.9) | 84 (49.1) | | |
| Others | 4 (30.8) | 9 (69.2) | | |
| **Religion** | | | | |
| Hindu | 130 (50.4) | 128 (49.6) | 1.008 | 0.315 |
| Other than Hindu | 34 (57.6) | 25 (42.4) | | |
| **Marital Status** | | | | |
| Married | 22 (41.5) | 31 (58.5) | 2.665 | 0.103 |
| Unmarried | 142 (53.8) | 122 (46.2) | | |
| **Living Arrangements** | | | | |
| Home | 133 (52) | 123 (48) | 0.025 | 0.874 |
| Other than home | 31 (50.8) | 30 (49.2) | | |
| **Family Type** | | | | |
| Nuclear Family | 43 (50.6) | 42 (49.4) | 0.061 | 0.805 |
| Other than nuclear | 121 (52.2) | 111 (47.8) | | |

Significant Level at 0.05.

the Turkish study's mean score of 31.69±6.91[14]. Most respondents (71.3%) have moderate stress, while 28.7% have high stress. This is similar to the study conducted in Pune, which found that 82.67% of participants had moderate stress, while 13.35% had high stress [18]. Still, it contrasts with the study conducted in Spain, which found that 47.92% of respondents had moderate stress [16]. Moderate stress in students was seen in most of the studies due to COVID-19 pandemic, lockdown and social distancing which was undertaken throughout the country to prevent spread of infection due devastating effect in health with increasing mortality and morbidity seen in many other countries [26]. As nursing education includes more clinical practice than theory classes which was disturbed due to lockdown, students faced difficulty in understanding the classes online without demonstration in clinical setting which also might be the factor to increase stress among them.

The brief coping score ranged from 28–105 with a Mean±SD of 65.55±*12.45*. The scores for problem-focused ranged from 8 to 32 with a Mean±SD of 21.69± 4.62, emotion-focused ranged from 12 to 47 with a Mean±SD of 27.9± 6.29, and avoidant coping ranged from 8 to 29 with a Mean±SD of 15.95± 4, which is in contrast to the findings of Pune where the mean score were 16.57±4.14, 16.87±4.16 and 9.44±2.64 respectively. More than half of the respondents (51.7%) have low coping, near about half (48.3%) of the respondents have high coping, which is in contrast to the study conducted in Pune where only 4.9% of participants had low coping, and 18.5% of participants had high coping [18].The present study's most commonly used coping strategy is problem-focused, similar to those conducted in Spain and Saudi [16,17].

There was no significant association between stress level, socio-demographic information, family-related information, and profession-related information of respondents, which is supported by a study done in Pune.There was no significant association between the level of

**Table 10. Association between level of coping and family related variables n = 317.**

| Variables | Level of Coping Strategies | | χ² | p-value |
|---|---|---|---|---|
| | Low No. (%) | High No. (%) | | |
| **Income** | | | | |
| 25000 and below | 27 (54) | 23 (46) | 0.134 | 0.935 |
| 25001 to 50000 | 56 (50.9) | 54 (49.1) | | |
| 50001 and above | 81 (51.6) | 76 (48.4) | | |
| **Father's Education** | | | 3.400 | 0.183 |
| Primary level and below | 38 (62.3) | 23 (37.7) | | |
| Secondary and higher secondary | 88 (48.9) | 92 (51.1) | | |
| Bachelor and above | 38 (50) | 38 (50) | | |
| **Mother's Education** | | | | |
| Primary level and below | 69 (50.7) | 67 (49.3) | 0.106 | 0.949 |
| Secondary and higher secondary | 79 (52.3) | 72 (47.7) | | |
| Bachelor and above | 16 (53.3) | 14 (46.7) | | |
| **Father's Occupation** | | | | |
| Agriculture | 16 (41) | 23 (59) | 5.097 | 0.165 |
| Sales and service/professional | 48 (49.5) | 49 (50.5) | | |
| Business | 94 (57) | 71 (43) | | |
| Others | 6 (37.5) | 10 (62.5) | | |
| **Mother's Occupation** | | | | |
| Homemaker | 123 (57.7) | 90 (42.3) | 14.227 | 0.003* |
| Agriculture | 6 (26.1) | 17 (73.9) | | |
| Sales and service | 30 (40.5) | 44 (59.5) | | |
| Others | 5 (71.4) | 2 (28.6) | | |

*Significant Level at 0.05 * Likelihood ratio.*

coping and socio-demographic information and profession-related information of respondents. Still, there was significant association between level of coping and mother's occupation (p = 0.003). Still, there was a significant association between the level of dealing with the age of respondents, year of study, and place of residents with a p-value <0.00 [18]. The findings of this study showed that those students whose mother is homemaker and takes care of her children coped well with the pandemic. The coping strategies adopted by the students would help them not only in present COVID-19 pandemic but have outlasting effects in future too.

The present study found relationship between stress and coping strategies adopted by respondents (r = 0.256, p = 0.001). The strength of the relationship was moderately positive, which is in contrast to the study conducted in Malaysia, which showed a significant negative correlation (r = -0.57, p<0.05) [27].

**Table 11. Relationship between perceived stress and coping strategies n = 317.**

| Variables | Correlation | p-value |
|---|---|---|
| Stress and Coping Score | 0.256 | 0.001 |

Significant Level at 0.05.

## Conclusion

It is concluded that most respondents have moderate stress, and almost one-third of the respondents have high stress. There were no significant influencing variables for the level of stress. More than half of the respondents had low coping, and nearly half had high coping. Among the three coping strategies, mostly adopted coping strategy problem-focused coping. The significant influencing variable for the level of coping was the mother's occupation that is respondents whose mother were home maker had high coping. A moderately positive relationship was found between stress and coping score, which is statistically significant. This depicts that the respondents with stress have high coping and vice versa. This finding points out a need to introduce stress management programs to help foster healthy coping skills among nursing students.

## Supporting information

**S1 Data.** . . .\Desktop\Original Data in SPSS.
(SAV)

## Acknowledgments

Researcher and the team are grateful to Yeti Health Science Academy for permission, support and encouragement toconduct the study. Last but not the least researcher is thankful to all data collection settings and all the participants.

## Author Contributions

**Conceptualization:** Sarmila Koirala, Rupa Devi Thapa, Sagun Bhandari, Alisha Rijal, Bhawani Shahi Thakuri.

**Data curation:** Sarmila Koirala, Rupa Devi Thapa, Alisha Rijal, Bhawani Shahi Thakuri.

**Formal analysis:** Sarmila Koirala, Alisha Rijal, Pooja Gauro.

**Funding acquisition:** Sarmila Koirala, Sagun Bhandari, Bhawani Shahi Thakuri.

**Investigation:** Sarmila Koirala, Pooja Gauro.

**Methodology:** Sarmila Koirala, Rupa Devi Thapa, Alisha Rijal, Pooja Gauro.

**Project administration:** Sarmila Koirala.

**Resources:** Sarmila Koirala.

**Software:** Sarmila Koirala, Alisha Rijal, Pooja Gauro.

**Supervision:** Sarmila Koirala, Rupa Devi Thapa.

**Validation:** Sarmila Koirala.

**Visualization:** Sarmila Koirala.

**Writing – original draft:** Sarmila Koirala, Bhawani Shahi Thakuri.

**Writing – review & editing:** Sarmila Koirala, Pooja Gauro.

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
