## [Decision Letter · Decision Letter 0]

2 Nov 2023

PONE-D-23-24440PERCEIVED STRESS AND COPING STRATEGIES AMONG NURSING STUDENTS TOWARDS REJOINING COLLEGE AFTER COVID-19PANDEMICPLOS ONE

Dear Dr. Koirala,

Thank you for submitting your manuscript to PLOS ONE. After careful consideration, we feel that it has merit but does not fully meet PLOS ONE’s publication criteria as it currently stands. Therefore, we invite you to submit a revised version of the manuscript that addresses the points raised during the review process.

We look forward to receiving your revised manuscript.

Kind regards,

Hariom Kumar Solanki, M.D.

Academic Editor

PLOS ONE

“Sarmila Koirala : Principal Investigator had role in application for fund, proposal writing, tool selection, data collection, analysis and preparing research report and manuscript.

Rupa Devi Thapa: Data collection and preparing research report and manuscript.

Sagun Bhandari: Writing proposal  and research report.

Alisha Rijal: Data collection and analysis

Bhawani Shahi Thakuri: Proposal writing, data collection and writing research report.

Pooja Gauro: Data collection, analysis and preparation of manuscript.”

“Researcher and the team are grateful to University Grants Commission for providing the fund and Yeti Health Science Academy for allowing to  conduct the study. Last but not the least research is thankful to data collection setting and all the participants.”

“Authors received grant form University Grant Commission for conducting this research. Fund was not sufficient so some fund was contributed by researchers and completed the study.

Authors who received award are:

Sarmila Koirala, Sagun Bhandari and Bhawani Shahi Thakuri.

Grant awarded is nepali rupees  Rs.100,000.

Webside of funder is : https://www.ugcnepal.edu.np/

Reviewers' comments:

Reviewer's Responses to Questions

**Comments to the Author**

1. Is the manuscript technically sound, and do the data support the conclusions?

Reviewer #1: Yes

Reviewer #2: Yes

2. Has the statistical analysis been performed appropriately and rigorously? 

Reviewer #1: No

Reviewer #2: Yes

3. Have the authors made all data underlying the findings in their manuscript fully available?

Reviewer #1: Yes

Reviewer #2: Yes

4. Is the manuscript presented in an intelligible fashion and written in standard English?

Reviewer #1: No

Reviewer #2: Yes

5. Review Comments to the Author

Reviewer #1: This is a well-written and informative manuscript that presents the results of a study on the perceived stress and coping strategies of nursing students returning to college after the COVID-19 pandemic. The study is well-designed and conducted, and the results are important and timely.

Specific Comments

The introduction provides a good overview of the background literature and the significance of the study. However, it could be improved by including a more explicit statement of the research questions.

The methodology section is clear and concise. The authors have adequately described the study design, sampling, data collection, and data analysis procedures.

The results section is well-organized and easy to follow. The authors have presented the results in a clear and concise manner, using tables and figures to summarize the key findings.

The discussion section is well-written and informative. The authors have discussed the implications of their findings for nursing education and practice. However, they could improve the discussion by more explicitly linking their findings to the existing literature.

Recommendations

The authors should include a more explicit statement of the research questions in the introduction.

The authors should more explicitly link their findings to the existing literature in the discussion.

Additional Comments

I have no concerns about dual publication, research ethics, or publication ethics.

Conclusion

This is a well-written and informative manuscript that presents the results of an important study on the perceived stress and coping strategies of nursing students returning to college after the COVID-19 pandemic. I recommend publication in a peer-reviewed journal after the authors have made the suggested revisions.

Reviewer #2: My dear authors

Many thanks for this well written manuscript and the value information regarding this topic

1- i just ask regarding table 1 you write at table 1 at socioeconomic information at categories of age ( age group below 19 years equal 7 but at descriptive information regarding this table as mentioned at bottom of table you write from 19 years and not mentioned the age group below 19

2- No DOI at all references

6. PLOS authors have the option to publish the peer review history of their article (what does this mean?). If published, this will include your full peer review and any attached files.

Reviewer #1: No

Reviewer #2: **Yes: **amr ahmed

---

## [Author Response · Author response to Decision Letter 0]

19 Dec 2023

First of all I would like to thank you so much for acknowledging our work and guiding us to make it a better manuscript. We all have tried to make corrections as per your suggestions. 

1. The editor of this journal has suggested and made some corrections to make the abstract short but informative. Corrections were made as per instructions. As English is not our native language so the language of manuscript and words picked were not satisfactory we are so happy to see the corrections made.

2. PLOS ONE style requirements for manuscript submission was reviewed and applied.

3. Grant information in the funding information section was checked but couldnot include the name of Sagun Bhandari and Bhawani Shahi Thakuri as three of us are fund recipient. In financial disclosure section the amount of fund received was mentioned but the received amount was not sufficient to complete the research so remaining fund was contributed by researchers self. Received amount from UGC: Rs.100,000. Total Expenses Rs.1,62,175.

4. The authors have declared that no competing interests exist in cover letter.

5. Funding information from acknowledgement was removed. As there is policy that we should acknowledge the funding agency during publication I have mentioned the name of UGC in acknowledgement section.

6. Data file was attached.

7. Reference list was reviewed and corrected.

Response to Author 1 Recommendations

1. Research questions were added in last part of the introduction portion.

2. Discussion section was revised to relate with existing literature.

 Response to Author 2 Recommendations

1. Age section was open ended question in questionnaire. Later range was created after calculating mean value of age and minimum and maximum age. Though the range first was mentioned as ≤ 19 years minimum age was 19 years and there were no one below 19 years so in description less than 19 years was not mentioned. To support this minimum and maximum age was kept in italic words in table with mean age.

2.DOI of some articles were not available and some references were newspaper, report from government so to support this URL/website where these information can be found are attached. Revisions on references were made as per feedback.

---

## [Decision Letter · Decision Letter 1]

5 Feb 2024

PONE-D-23-24440R1PERCEIVED STRESS AND COPING STRATEGIES AMONG NURSING STUDENTS TOWARDS REJOINING COLLEGE AFTER COVID-19PANDEMICPLOS ONE

Dear Dr. Koirala,

Thank you for submitting your manuscript to PLOS ONE. After careful consideration, we feel that it has merit but does not fully meet PLOS ONE’s publication criteria as it currently stands. Therefore, we invite you to submit a revised version of the manuscript that addresses the points raised during the review process.

We look forward to receiving your revised manuscript.

Kind regards,

Hariom Kumar Solanki, M.D.

Academic Editor

PLOS ONE

Journal Requirements:

Reviewers' comments:

Reviewer's Responses to Questions

**Comments to the Author**

1. If the authors have adequately addressed your comments raised in a previous round of review and you feel that this manuscript is now acceptable for publication, you may indicate that here to bypass the “Comments to the Author” section, enter your conflict of interest statement in the “Confidential to Editor” section, and submit your "Accept" recommendation.

Reviewer #1: All comments have been addressed

Reviewer #2: All comments have been addressed

Reviewer #3: (No Response)

2. Is the manuscript technically sound, and do the data support the conclusions?

Reviewer #1: Partly

Reviewer #2: Yes

Reviewer #3: Partly

3. Has the statistical analysis been performed appropriately and rigorously? 

Reviewer #1: Yes

Reviewer #2: I Don't Know

Reviewer #3: Yes

4. Have the authors made all data underlying the findings in their manuscript fully available?

Reviewer #1: Yes

Reviewer #2: Yes

Reviewer #3: Yes

5. Is the manuscript presented in an intelligible fashion and written in standard English?

Reviewer #1: Yes

Reviewer #2: Yes

Reviewer #3: No

6. Review Comments to the Author

Reviewer #1: Overall, the articles well-written and effectively introduces the research topic and its significance. Implementing the suggested improvements could further strengthen its clarity and impact.

Reviewer #2: My dear authors;

Many thanks for the prompt response for updating all comments and fixed it

with my best wishes

Reviewer #3: I would like to thank the editors for providing me the opportunity to review the article.

General comments:

There is still a need for improvising the text in context of english language.

Comments to authors:

1. Introduction

1a.Introduction can be concised to 3 paragraphs, last parah describing the rationale and objective of the study.

2. Methodology:

2a: Data collection procedure needs more elaboration like: personnel involved in data collection, how the calculated sample size was achieved and were selected during data collection.

2b: Research instrument description, should also include the duration period considered for assessing the perceived stress and coping strategies.

2c: Authors need to mention, the operational definitions of variables such as type of family, on what basis occupation, education and income was classified.

3. Results:

There is requirement of merging of tables.

Table 3.1, 3.2 & 3.3 can be merged

Similarly, Tables 3.4,3.5; 3.6 to be merged.

Also, Tables 3.7, 3.8,3.9,3.10 require merging. Perceive stress and coping strategies can be put in separate columns under same table.

There is no requirement of table 3.11, simply can be highlighted in text of results section.

4. Discussion:

The authors have mentioned in the last para of discussion that they have found significant relationship between stress and coping strategies was only 0.256, as it is only moderately correlated. Kindly modify the sentence.

5. Conclusion:

“ The significant influencing variables for level of coping was mother’s occupation; however the classification of occupation of mother needs further consideration.

5. Conclusion :

7. PLOS authors have the option to publish the peer review history of their article (what does this mean?). If published, this will include your full peer review and any attached files.

Reviewer #1: **Yes: **NUHA AMER Al-Aghbari

Reviewer #2: **Yes: **Amr Ahmed

Reviewer #3: No

---

## [Author Response · Author response to Decision Letter 1]

17 Mar 2024

Response to Author 1 Recommendations

Thank you so much for appreciation and guidance.

Response to Author 2 Recommendations

Thank you so much for appreciation and guidance.

Response to Author 3 Recommendations

Thank you so much for constructive feedback.

1. Reviewer suggested to concise introduction in 3 paragraphs .we tried to concise in three but I am so sorry to say that we are able to concise the introduction in 4 paragraphs.

2. In methodology personnel involved in data collection was added in data collection procedure. How calculated sample was achieved and selected was added in sampling section.

3. Duration for assessing perceived stress and coping was added in instrumentation section.

4. There was no operational definition section in this journal guidelines so it was not mentioned earlier. But as per reviewer comment operational definitions were added in 1.4. Operational definitions are defined as per our country's context.

5. Reviewer has suggested to merge some tables and remove table 3.11. We appreciate your comments but want to clear that tables are made according to objectives so to clarify this objectives were added in manuscript. For independent variables there were socio-demographic variables, family related variables and profession related variables so these are shown separately. To identify level of coping and level of stress were separate objectives so these are shown separately and same with association. If it is mandatory to merge tables please let us know we will merge them.

6. In discussion section term significant relationship was replaced with only relationship and this relation was explained in next sentence which was moderately positive.

7. In conclusion section it was made clear how mother's occupation influenced level of coping.

I hope we have answered the queries and made corrections as per need. Please let us know if more corrections are needed or still there remains unanswered queries.

---

## [Decision Letter · Decision Letter 2]

2 Jun 2024

PERCEIVED STRESS AND COPING STRATEGIES AMONG NURSING STUDENTS TOWARDS REJOINING COLLEGE AFTER COVID-19PANDEMIC

PONE-D-23-24440R2

Dear Dr. Koirala,

We’re pleased to inform you that your manuscript has been judged scientifically suitable for publication and will be formally accepted for publication once it meets all outstanding technical requirements.

Kind regards,

Hariom Kumar Solanki, M.D.

Academic Editor

PLOS ONE

Additional Editor Comments (optional):

Reviewers' comments:

Reviewer's Responses to Questions

**Comments to the Author**

1. If the authors have adequately addressed your comments raised in a previous round of review and you feel that this manuscript is now acceptable for publication, you may indicate that here to bypass the “Comments to the Author” section, enter your conflict of interest statement in the “Confidential to Editor” section, and submit your "Accept" recommendation.

Reviewer #1: All comments have been addressed

Reviewer #4: (No Response)

2. Is the manuscript technically sound, and do the data support the conclusions?

Reviewer #1: Yes

Reviewer #4: (No Response)

3. Has the statistical analysis been performed appropriately and rigorously? 

Reviewer #1: I Don't Know

Reviewer #4: (No Response)

4. Have the authors made all data underlying the findings in their manuscript fully available?

Reviewer #1: Yes

Reviewer #4: (No Response)

5. Is the manuscript presented in an intelligible fashion and written in standard English?

Reviewer #1: Yes

Reviewer #4: (No Response)

6. Review Comments to the Author

Reviewer #1: "The authors have comprehensively addressed all the comments raised. No further revisions are necessary from my end."

Reviewer #4: (No Response)

7. PLOS authors have the option to publish the peer review history of their article (what does this mean?). If published, this will include your full peer review and any attached files.

Reviewer #1: **Yes: **NUHA AMER ABDULWAHAB AL-AGHBARI

Reviewer #4: No

---

## [Editor Report · Acceptance letter]

24 Jun 2024

PONE-D-23-24440R2 

PLOS ONE

Dear Dr. Koirala, 

I'm pleased to inform you that your manuscript has been deemed suitable for publication in PLOS ONE. Congratulations! Your manuscript is now being handed over to our production team.

Kind regards, 

on behalf of

Dr. Hariom Kumar Solanki 

Academic Editor

PLOS ONE